# Immunogenicity, reactogenicity, and IgE-mediated immune responses of a mixed whole-cell and acellular pertussis vaccine schedule in Australian infants: A randomised, double-blind, noninferiority trial

Gladymar Pérez Chacón[1,2], Marie J. Estcourt[3], James Totterdell[3], Julie A. Marsh[1], Kirsten P. Perrett[4,5], Dianne E. Campbell[6,7], Nicholas Wood[7,8], Michael Gold[9], Claire S. Waddington[10], Michael O' Sullivan[11,12,13], Sonia McAlister[1,12], Nigel Curtis[5,14,15], Mark Jones[3], Peter B. McIntyre[16], Patrick G. Holt[17], Peter C. Richmond[1,12], Tom Snelling[1,2,3]*

1 Wesfarmers Centre of Vaccines and Infectious Diseases, Telethon Kids Institute, Nedlands, Western Australia, Australia, 2 School of Population Health, Faculty of Health Science, Curtin University, Perth, Western Australia, Australia, 3 Sydney School of Public Health, Faculty of Medicine and Health, University of Sydney, Sydney, New South Wales, Australia, 4 Royal Children's Hospital, Murdoch Children's Research Institute, Parkville, Victoria, Australia, 5 Department of Paediatrics, The University of Melbourne, Melbourne, Victoria, Australia, 6 Department of Allergy and Immunology, The Children's Hospital at Westmead, Sydney, New South Wales, Australia, 7 Discipline of Child and Adolescent Health, The University of Sydney, Sydney, New South Wales, Australia, 8 The Children's Hospital at Westmead, Sydney, New South Wales, Australia, 9 Discipline of Paediatrics, School of Medicine, University of Adelaide, Adelaide, South Australia, Australia, 10 Department of Medicine, University of Cambridge, Cambridge, United Kingdom, 11 Department of Immunology, Perth Children's Hospital, Nedlands, Western Australia, Australia, 12 Division of Paediatrics, School of Medicine, The University of Western Australia, Perth, Western Australia, Australia, 13 Telethon Kids Institute, Nedlands, Western Australia, Australia, 14 Infectious Diseases Unit, Royal Children's Hospital Melbourne, Parkville, Victoria, Australia, 15 Infectious Diseases Group, Murdoch Children's Research Institute, Parkville, Victoria, Australia, 16 Dunedin School of Medicine, University of Otago, Dunedin, New Zealand, 17 Wal-yan Respiratory Research Centre, Telethon Kids Institute, University of Western Australia, Nedlands, Western Australia, Australia

* tom.snelling@sydney.edu.au

**Data Availability Statement:** Data from the study will be available at the completion of follow-up,

## Abstract

### Background

In many countries, infant vaccination with acellular pertussis (aP) vaccines has replaced use of more reactogenic whole-cell pertussis (wP) vaccines. Based on immunological and epidemiological evidence, we hypothesised that substituting the first aP dose in the routine vaccination schedule with wP vaccine might protect against IgE-mediated food allergy. We aimed to compare reactogenicity, immunogenicity, and IgE-mediated responses of a mixed wP/aP primary schedule versus the standard aP-only schedule.

### Methods and findings

OPTIMUM is a Bayesian, 2-stage, double-blind, randomised trial. In stage one, infants were assigned (1:1) to either a first dose of a pentavalent wP combination vaccine (DTwP-Hib-

analysis and reporting of OPTIMUM stage two; no end date. Deidentified, individual participant data and a data dictionary defining each field in the set, will be made available to researchers who provide a methodologically sound proposal to the University of Sydney, Australia (Ms Nicole Wong – Clinical Trial Support Lead; Department: Pro Vice-Chancellor, Research; nicole.wong@sydney.edu.au), subject to a signed data access agreement and any necessary ethics approvals (see ANZCTR trial registration). The study protocol and statistical analysis plan are published, other study related documentation is available on request.

**Funding:** This is an investigator-initiated study supported by grants from the Australian National Health and Medical Research Council (NHMRC; GNT 1158722; https://nhmrc.gov.au) awarded to TS, DEC, MG, JAM, PCR, NW, KPP, and Telethon New Children's Hospital Research Fund 2012 (Round 1; https://research-repository.uwa.edu.au/en/projects/telethon-new-childrens-hospital-research-fund-the-safety-and-a) awarded to TS, PCR, and PGH. GPC was supported by a Stan Perron Post-PhD Career Launching Award (2022; https://perronfoundation.org.au), the Australian Department of Education and Training Endeavour Scholarship (https://internationaleducation.gov.au/scholarships/Pages/Endeavour-Leadership-Program.aspx), and top-up scholarships from the Wesfarmers Centre of Vaccine and Infectious Diseases at the Telethon Kids Institute (https://infectiousdiseases.telethonkids.org.au) and the Forrest Research Foundation (https://forrestresearch.org.au/). KPP is supported by a NHMRC fellowship, GNT2008911 and a Melbourne Children's Clinician-Scientist Fellowship awarded by the University of Melbourne (https://www.unimelb.edu.au/), Murdoch Children's Research Institute (https://www.mcri.edu.au/), and the Royal Children's Hospital Foundation (https://www.rchfoundation.org.au/). TS is supported by a Medical Research Future Fund Investigator Grant (MRF119515; https://health.gov.au/our-work/medical-research-future-fund). The funders had no role in the design and conduct of this trial, in the analyses of the data, or in the decision to submit the results for publication.

**Competing interests:** KPP has received research grants from Aravax, DBV Technologies, Novartis and Siolta and consultant fees from Aravax, paid to their institution, outside the submitted work. Unrelated to the work presented in this paper, DEC declared part-time employment in DBV Technologies, stock and stock options of DBV Technologies, and an honorarium for the AllerGenis Advisory Board. Unrelated to the work

HepB, Pentabio PT Bio Farma, Indonesia) or a hexavalent aP vaccine (DTaP-Hib-HepB-IPV, Infanrix hexa, GlaxoSmithKline, Australia) at approximately 6 weeks old. Subsequently, all infants received the hexavalent aP vaccine at 4 and 6 months old as well as an aP vaccine at 18 months old (DTaP-IPV, Infanrix-IPV, GlaxoSmithKline, Australia). Stage two is ongoing and follows the above randomisation strategy and vaccination schedule. Ahead of ascertainment of the primary clinical outcome of allergist-confirmed IgE-mediated food allergy by 12 months old, here we present the results of secondary immunogenicity, reactogenicity, tetanus toxoid IgE-mediated immune responses, and parental acceptability endpoints. Serum IgG responses to diphtheria, tetanus, and pertussis antigens were measured using a multiplex fluorescent bead-based immunoassay; total and specific IgE were measured in plasma by means of the ImmunoCAP assay (Thermo Fisher Scientific). The immunogenicity of the mixed schedule was defined as being noninferior to that of the aP-only schedule using a noninferiority margin of 2/3 on the ratio of the geometric mean concentrations (GMR) of pertussis toxin (PT)-IgG 1 month after the 6-month aP. Solicited adverse reactions were summarised by study arm and included all children who received the first dose of either wP or aP. Parental acceptance was assessed using a 5-point Likert scale. The primary analyses were based on intention-to-treat (ITT); secondary per-protocol (PP) analyses were also performed. The trial is registered with ANZCTR (ACTRN12617000065392p).

Between March 7, 2018 and January 13, 2020, 150 infants were randomised (75 per arm). PT-IgG responses of the mixed schedule were noninferior to the aP-only schedule at approximately 1 month after the 6-month aP dose [GMR = 0·98, 95% credible interval (0·77 to 1·26); probability (GMR > 2/3) > 0·99; ITT analysis]. At 7 months old, the posterior median probability of quantitation for tetanus toxoid IgE was 0·22 (95% credible interval 0·12 to 0·34) in both the mixed schedule group and in the aP-only group. Despite exclusions, the results were consistent in the PP analysis. At 6 weeks old, irritability was the most common systemic solicited reaction reported in wP (65 [88%] of 74) versus aP (59 [82%] of 72) vaccinees. At the same age, severe systemic reactions were reported among 14 (19%) of 74 infants after wP and 8 (11%) of 72 infants after aP. There were 7 SAEs among 5 participants within the first 6 months of follow-up; on blinded assessment, none were deemed to be related to the study vaccines. Parental acceptance of mixed and aP-only schedules was high (71 [97%] of 73 versus 69 [96%] of 72 would agree to have the same schedule again).

## Conclusions

Compared to the aP-only schedule, the mixed schedule evoked noninferior PT-IgG responses, was associated with more severe reactions, but was well accepted by parents. Tetanus toxoid IgE responses did not differ across the study groups.

## Trial registration

Trial registered at the Australian and New Zealand Clinical 207 Trial Registry (ACTRN12617000065392p).

presented in this paper, MO is the non-remunerated Director of ASCIA, a not-for-profit company and the peak professional body for allergy/immunology specialists in Australia and New Zealand. Unrelated to the work presented in this paper, PCR has served on pertussis vaccine scientific advisory boards for GlaxoSmithKline and Sanofi on behalf of his institution. He participated in multicentre vaccine trials of pertussis vaccines sponsored by industry, also unrelated to the work presented in this paper. He has received no personal remuneration for these activities. These organisations/industries had no role in the study design, data collection, data analysis, data interpretation, or writing of this report.

**Abbreviations:** aP, acellular pertussis; DT, diphtheria toxoid; FHA, filamentous haemagglutinin; FIM 2/3, fimbriae 2/3; GMR, geometric mean ratio; HepB, hepatitis B; Hib, *Haemophilus influenzae* type b; IPV, inactivated poliovirus vaccine; ITT, intention-to-treat; OPTIMUM, Optimising Immunisation Using Mixed Schedules; PP, per-protocol; PT, pertussis toxin; RCT, randomised controlled trial; SAE, serious adverse event; TT, tetanus toxoid; WHO, World Health Organization; wP, whole-cell pertussis; 13vPCV, 13-valent pneumococcal conjugate vaccine.

## Author summary

### Why was this study done?

- Evidence-based strategies to lower the risk of food allergies developing in young children are limited.

- A previous observational study suggested that food allergies appear to be less common among Australian-born children vaccinated with a first dose of whole-cell whooping cough (wP) vaccine before 4 months old versus those receiving a first dose of acellular whooping cough (aP) vaccine at the same age.

### Why did the researchers do and find?

- In the first stage of a 2-stage, double-blind, noninferiority trial, Australian-born infants were randomised to receive a first dose of wP versus aP at approximately 6 weeks old; in both study groups, aP vaccine is given at 4 and 6 months old.

- Stage one examined the immunogenicity, reactogenicity, and IgE-mediated immune responses of the mixed vaccine schedule (wP/aP/aP) versus the standard aP-only vaccination strategy (aP/aP/aP) in a cohort of 150 infants; stage two is ongoing and was designed to ascertain the development of food allergy by 12 months old (primary outcome) in a cohort of up to 3,000 infants.

- At 7 months old, the mixed schedule was noninferior to the standard aP-only strategy with respect to pertussis toxin IgG responses; at the same age, we observed no difference across the study groups in IgE responses to hen's egg and tetanus toxoid antigens.

- The mixed schedule was more reactogenic but well accepted by parents.

### What do these findings mean?

- These findings support the acceptable immunogenicity and reactogenicity of the mixed schedule and are relevant for countries in which wP and aP vaccines are licensed and readily available.

- While the reported IgE responses are not conclusive, further studies of CD4+ T cell polarisation in response to pertussis antigens, along with primary outcome data will provide a comprehensive picture of the atopic immunophenotypic responses across the study groups.

- Additional evidence is required to understand the population-level acceptability of the mixed vaccination strategy.

## Introduction

Whole-cell pertussis (wP) vaccines cause adverse reactions in infants frequently, but these are generally benign and self-limited [1]. Since the early 1990s, wP vaccine formulations have been discontinued from use in most high- and some middle-income countries and replaced by less

reactogenic subunit acellular (aP) vaccines [2]. Data from pertussis outbreaks suggest that primary schedules comprising at least 1 wP vaccine dose provide better long-term protection than aP-only primary schedules [3,4].

We have previously reported that compared to matched population controls, Australian children with IgE-mediated food allergy were less likely to have received wP rather than aP as their first pertussis vaccine dose in infancy (OR 0·77; 95% CI 0·62 to 0·95) [5]. In the same vein, a population-based cohort study of nearly 220,000 Australian children found that a first dose of wP vaccine before 4 months old, rather than aP vaccine, was associated with 53% lower risk of hospitalisations for food-induced anaphylaxis between 5 and 15 years old (adjusted hazard ratio 0.47, 95% CI 0.26 to 0.83) [6]. In contrast with these findings, however, we found no evidence of a difference in admissions to hospital coded as asthma during the same study period (adjusted hazard ratio 1.02; 95% CI 0.94 to 1.12) [7].

Previous studies have described differences in the immune effects arising from vaccination with wP versus aP vaccines in infancy. Compared to exclusive vaccination with aP, which elicits a T-helper $(Th)_2$-polarised immunophenotype, a first dose of wP induces relative $Th_1/Th_{17}$ immune polarisation, reducing the production of aP antigen-specific type 2 cytokines and down-regulating the synthesis of total IgE, and specific IgE responses against tetanus toxoid (TT), diphtheria toxoid (DT), pertussis toxin (PT), and food antigens [8–11]. Moreover, enhanced PT-IgE and PT-$IgG_4$ responses to filamentous haemagglutinin (FHA) and fimbriae 2/3 (FIM 2/3) after dTap boosting have been described in a subset of adults who had received primary infant vaccination with aP only doses, but not among those primed with a homologous wP schedule [12]. We, therefore, hypothesise that mixed priming using a first dose of wP followed by aP vaccine doses could protect against IgE-mediated food allergy, while offering noninferior protection against pertussis and an acceptable reactogenicity profile. To test this, we are conducting a 2-stage randomised controlled trial (RCT) to assess clinical, immunological, and safety endpoints in Australian infants vaccinated with a mixed (wP/aP/aP) primary vaccine schedule, compared with the standard aP-only (aP/aP/aP) vaccine schedule. Ahead of ascertainment of the primary clinical outcome of IgE-mediated food allergy by 12 months old, here we present the immunogenicity and reactogenicity outcomes, which were assessed in the first 150 infants enrolled in Perth, Western Australia (WA), at approximately 6 and 7 months old, as well as total IgE, TT, and food antigen-specific IgE responses at the same ages.

## Methods

### Study design and participants

The OPTIMUM (Optimising Immunisation Using Mixed Schedules) study is a Bayesian group-sequential, 2-stage, multicentre, randomised, parallel group, double-blind controlled trial with an adaptive design. Stage one was designed to obtain detailed solicited reactogenicity data following each pertussis vaccine dose and post-priming immune response data for the first 150 enrolled infants. Stage two was designed to assess the primary endpoint of IgE-mediated food allergy, with less intensive follow-up, but with sufficient sample size (up to 3,000 infants, including the first 150 enrollees in stage one) to provide appropriate levels of statistical power. The analyses that we describe here involve the first 6 months of follow-up of the first 150 WA-born infants enrolled in stage one (Perth Children's Hospital/Telethon Kids Institute, WA). The eligibility criteria, enrolment methods, and visit schedule are provided in the protocol (https://doi.org/10.1136/bmjopen-2020-042838) and statistical analysis plan (https://doi.org/10.1186/s13063-021-05874-6) [13,14]. Eligible participants were healthy infants aged between 6 and <12 weeks old and born after 32 weeks' gestation. Parents of each participating

infant provided written informed consent. The trial was approved by the Child and Adolescent Health Service Ethics Committee, WA, Australia (RGS 00019).

## Randomisation and masking

Randomisation was by computer-generated allocation sequence prepared by the trial statistician and based on randomly permuted blocks of size 6, 8, or 10. Infants were assigned in a 1:1 ratio to receive either the intervention (pentavalent wP combination vaccine: DT, TT, wP, *Haemophilus influenzae* type b [Hib], and hepatitis B [HepB] vaccine; DTwP-Hib-HepB, Pentabio, PT Bio Farma, Indonesia) or the comparator (hexavalent aP combination vaccine, which includes inactivated poliovirus vaccine [IPV] types 1, 2, and 3 in its formulation: DTaP-Hib-HepB-IPV, Infanrix hexa, GlaxoSmithKline, Australia) at approximately 6 weeks old [15–17]. An unblinded research nurse obtained the next sequential vaccine allocation and prepared it into a syringe that was covered with an opaque label. Following vaccination, this nurse had no involvement in subsequent study procedures. Parents of children in the stage one cohort were unblinded in May 2023, after the last child completed the study.

## Procedures

At approximately 6 weeks old ("Day 0"), an intramuscular dose (0.5 mL) of either a World Health Organization (WHO)-prequalified pentavalent wP combination vaccine or a hexavalent aP combination vaccine was administered into the anterolateral aspect of the right thigh. At approximately 4 and 6 months old, all participants received the standard (6-in-1) aP combination vaccine (as routinely recommended by the Australian immunisation schedule) at the same injection site (anterolateral aspect of the right thigh) [18]. The 13-valent pneumococcal conjugate vaccine (13vPCV) and monovalent rotavirus vaccine (RV1) were coadministered at approximately 6 weeks and 4 months old per Australian recommendations [18]. National guidelines also recommend all children receive at least 3 doses of IPV as part of their childhood schedule; to ensure this was achieved while preserving blinding, children assigned to both the mixed and the aP-only schedule received a dose of DTaP-IPV (Infanrix-IPV, GlaxoSmithKline, Australia) at 18 months old [18,19]. A prophylactic dose of paracetamol (15 mg/Kg) was administered immediately before the 6-week doses per Australian guidelines for wP vaccines. Two additional doses of paracetamol 6-hour apart were recommended, but not observed by the researchers.

Solicited systemic and local adverse reactions following pertussis primary vaccinations were ascertained once a day for 7 days and recorded by the participant's parent in a diary card. These were fever (axillary temperature $\geq 38°C$), irritability, restlessness, vomiting, diarrhoea, anorexia, drowsiness, as well as erythema (redness), swelling, induration (hardness), and pain at the injection site. Serious adverse events (SAEs) were defined as any adverse event/reaction that resulted in death, was life-threatening, required hospitalisation or prolongation of existing hospitalisation, resulted in persistent or significant disability or incapacity, or was a congenital anomaly, or birth defect [20]. The total number of SAEs and adverse events of special interest (breakthrough pertussis infections and hypotonic hyporesponsive episodes) occurring within the first 6 months of follow-up were reported. SAEs and unsolicited adverse events by treatment arm will be reported at the end of the trial. To ascertain overall satisfaction of the vaccines administered, parents were asked about their agreement with the statement: "I would be willing to vaccinate another child with the same combination of vaccines administered;" responses were recorded in diary cards on Day 6 using a 5-point Likert scale, from "strongly disagree" to "strongly agree".

Peripheral blood samples were obtained immediately before the 6-month aP dose and approximately 1 month afterwards. All immunological assays were performed blinded. Serum IgG responses to diphtheria, tetanus, and pertussis antigens were measured using a multiplex fluorescent bead-based immunoassay. The assay's operating characteristics and quality control procedures are described elsewhere [21,22]. The threshold for seropositivity was defined as concentrations ≥5 IU/mL (5,000 mIU/mL) for pertussis antigens (i.e., PT, FHA, pertactin [PRN], and FIM2/3) [23]. Long-term seroprotective concentrations for TT and DT were defined at ≥100 mIU/mL [24,25].

Total and specific IgE to hen's egg antigens and TT were measured in plasma by means of the ImmunoCAP assay (Thermo Fisher Scientific) using the limits of detection (LLoD) and quantitation (LLoQ) specified by the manufacturer. The analyses were performed at PathWest (Department of Health, WA), under standardised study operating procedures. The assay's operating characteristics are described elsewhere [26–28]. The total IgE concentrations were reported in the range of 0·1 kU/L to 100 kU/L for low-range assays or 2 to 5,000 kU/L if exceeded (accuracy of the immunoassay is within ±8% of the WHO IgE reference value with a 95% probability; trueness of the calibrator within ±2% compared to the WHO IgE reference value with a 95% probability) [27]. IgE concentrations to specific antigens were reported in the range of 0·00 $kU_a/L$ to 100·00 $kU_a/L$ [28]. The LLoD for antigen/allergen-specific IgE was ≥0·01 kU/L; the LLoQ for total and specific IgE was ≥0·1 $kU_{(a)}/L$ [28].

## Outcomes

Vaccine immunogenicity outcomes included IgG responses to PT, FHA, PRN, TT, and DT immediately before the 6-month aP dose and approximately 1 month later. Outcomes related to IgE-mediated immune responses included total IgE and specific IgE against hen's egg antigens and TT assessed immediately before the 6-month aP dose and approximately 1 month later. Reactogenicity and tolerability outcomes included the occurrence of specific solicited local and systemic adverse reactions in the 7 days following each scheduled dose, and the parent-reported acceptability of the doses administered.

## Statistical analysis

The statistical analysis plan was written by the trial statistician and approved by study investigators prior to unblinding of group allocations [14]. Intention-to-treat (ITT) and per-protocol (PP) estimands were considered for each statistical model. The ITT analysis set included all infants who received at least the first dose of wP or aP, irrespective of any subsequent deviations from the study protocol. The PP analysis set included only infants with outcomes measured without deviation from the protocol-specified vaccination and blood collection schedules. Missing outcomes were assumed to be missing-at-random. Based on previous clinical/immunological data [10], we determined that a sample size of 150 infants, allocated 1:1 to each schedule, would provide a power of at least 80% to detect a 20-percentage point decrease in the proportion of participants with detectable TT-specific-IgE at 7 months old between infants primed with mixed wP/aP and aP-only. This controls the significance level (probability of falsely declaring a difference if none existed) at 5% for a one-sided hypothesis test (wP/aP/aP less than aP-only) and assumes that the proportion of participants with detectable TT-IgE-specific-IgE at 7 months old is at least 25% in the aP-only arm. Immunogenicity and outcomes related to IgE-mediated immune responses were summarised by the sample GMCs as well as seropositivity rate at each time point. Each antigen/allergen type was analysed using Bayesian multivariate-normal linear regression models on the available $\log_{10}$ concentrations with an unstructured covariance matrix shared across both treatment groups. The geometric mean

ratio (GMR) of the mixed schedule relative to the aP-only schedule at each time point was estimated. A noninferiority margin of 2/3 on the GMR was used to assess the noninferiority of the mixed schedule versus the aP-only schedule with regard to PT-IgG approximately 1 month after the 6-month aP dose per WHO guidelines [29]. To estimate the models, IgE-specific values reported as $0.00$ kU$_a$/L were treated as left-censored at $0.005$ kU$_a$/L. All adjusted models included sex, birth order, breastfeeding status, delivery method, family history of atopic disease, and parental income as baseline covariates.

To supplement the IgG and IgE models of the raw concentrations, the event of IgG seropositivity for each vaccine antigen with respect to the specified threshold and IgE concentrations $\geq 0.01$ KU$_{(a)}$/L were analysed via Bayesian logistic mixed effects models. The models were used to derive standardised differences of the mixed schedule compared to the aP-only schedule for probability of seropositivity and IgE $\geq 0.01$ KU/L, respectively. The reactogenicity data were summarised by the distribution of each solicited adverse event for each vaccine occasion at the study site clinic and the highest/worst reaction grade experienced. Details for the assumed Bayesian models are given in Section A in S1 Text. Analyses were performed in R version $4.3.1$ using Stan version $2.33.1$ (via CmdStanR version $0.6.1$). This trial is overseen by an independent Data and Safety Monitoring Committee and registered at the Australian and New Zealand Clinical Trial Registry (ACTRN12617000065392p).

## Results

Between March 7, 2018 and January 13, 2020, 153 infants were screened for eligibility (Fig 1 and Section B, in S1 Text), of whom 150 were randomised to wP versus aP for their 6-week pertussis vaccine dose (75 in each group).

Baseline characteristics were well balanced across mixed and aP-only schedules (Table 1). Except for 3 infants in the mixed schedule arm, infants in this cohort were born from an aP-vaccinated mother in the preceding pregnancy.

### Immunogenicity

All available IgG responses are summarised in Figs 2 and 3 and in Sections C.1–C.5 in S1 Text; PP analyses are included in Sections C.6–C.9 in S1 Text. Results from unadjusted models were generally consistent with the adjusted models, and, therefore, unadjusted models are not reported.

At 6 months old, 72 (96%) of 75 infants in the aP-only schedule group and 68 (91%) of 75 infants in the mixed schedule group had immunogenicity outcome data available. At 7 months old, 69 (92%) of 75 infants in the aP-only schedule group and 67 (89%) 75 infants in the mixed schedule group had immunogenicity outcome data available.

At 7 months old, 68 (99%) of 69 infants in the aP-only group and 66 (99%) of 67 infants in the mixed schedule group had PT-, FHA-, and PRN-IgG concentrations above the specified seropositivity threshold, as well as TT- and DT-IgG above the specified seroprotective threshold (Section C.2 in S1 Text). In the mixed schedule group, infants had PT-IgG GMC of $4.94$ IU/mL at approximately 6 months old, which increased to $26.38$ IU/mL at approximately 7 months old. In the aP-only schedule group, infants had PT-IgG GMC of $13.66$ IU/mL at approximately 6 months old, which increased to $27.69$ IU/mL at approximately 7 months old (Section C.2 in S1 Text). The posterior median GMR for PT-IgG (mixed schedule/aP-only schedule) at 7 months old was $0.98$ (95% credible interval [CrI] $0.77$ to $1.26$; probability (GMR > 2/3) > $0.99$). A decrease in IgG GMCs for FIM 2/3 and an increase in IgG GMCs for FHA, PRN, DT, and TT were observed at 7 months compared to 6 months old for both study groups (Section C.2 in S1 Text). The remaining IgG GMR posterior summaries are included in Table 2. Analyses of IgG concentrations and

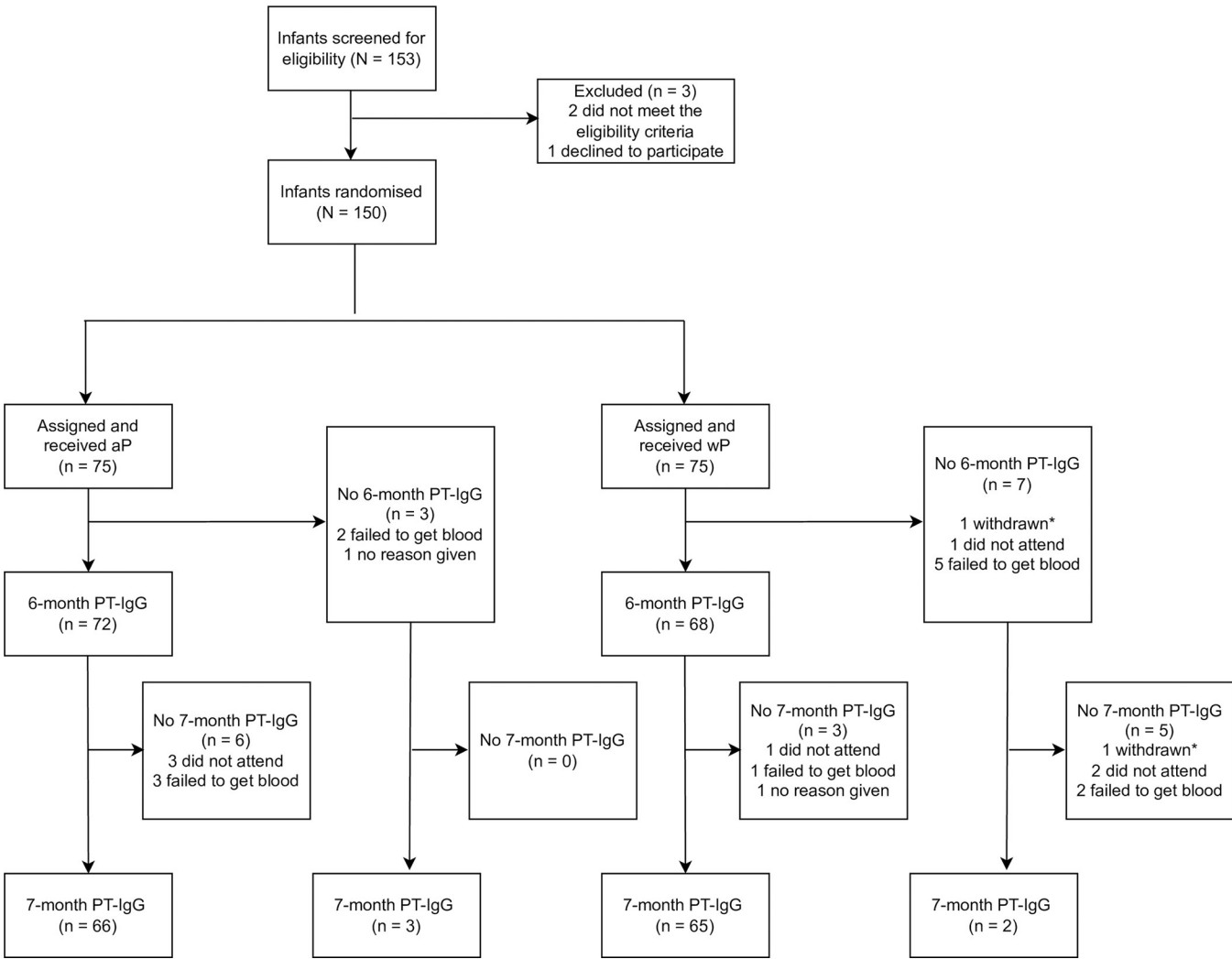

**Fig 1. Trial profile.** aP, acellular pertussis vaccine; PT, pertussis toxin; wP, whole-cell pertussis vaccine. *Refers to the same study participant.

seropositivity, as well as 4-fold rise in IgG concentrations from 6-month-old to 7-month-old are reported in Sections C.3–C.5 in S1 Text. Despite exclusions, the results were consistent in the PP analysis set (Sections C.6–C.9 in S1 Text).

## IgE-mediated immune responses

Total IgE and IgE responses to hen's egg and TT antigens are summarised in Fig 4 and in Sections D.1–D.5 in S1 Text; PP analyses are included in Sections D.6–D.9 in S1 Text. Results from unadjusted models were generally consistent with the adjusted models, and, therefore, unadjusted models are not reported.

At approximately 7 months old, total IgE levels $\geq 0.1$ kU/L were observed in all infants assigned to the mixed schedule (GMC = 7·71 kU/L) and to the aP-only schedule (GMC = 6·36 kU/L) who had available plasma samples (mixed schedule group: 57 [76%] of 75 infants; aP-only group: 66 [88%] of 75 infants; Section D.2 in S1 Text). The posterior median 7-month GMR for total IgE (mixed schedule /aP-only schedule) was 1·14 (95% CrI 0·75 to 1·72; probability (GMR < 1) 0·27; section D.3 in S1 Text). Summaries of the posterior probability of

**Table 1. Baseline and demographic characteristics.**

| | First dose of aP combination vaccine (*n* = 75) | First dose of wP combination vaccine (*n* = 75) | Overall (*n* = 150) |
|---|---|---|---|
| **History of atopic diseases in first degree relatives—n (%)** | | | |
| No | 16 (21) | 20 (27) | 36 (24) |
| Yes | 58 (77) | 54 (72) | 112 (75) |
| Unknown | 1 (1) | 1 (1) | 2 (1) |
| **Sex—n (%)** | | | |
| Female | 36 (48) | 38 (51) | 74 (49) |
| Male | 39 (52) | 37 (49) | 76 (51) |
| **Household income—n (%)** | | | |
| <$18,000 | 1 (1) | 0 (0) | 1 (1) |
| $18,000 to $37,000 | 0 (0) | 1 (1) | 1 (1) |
| $37,001 to $87,000 | 5 (7) | 2 (3) | 7 (5) |
| $87,001 to $180,000 | 34 (45) | 42 (56) | 76 (51) |
| >$180,000 | 35 (47) | 30 (40) | 65 (43) |
| **Maternal education—n (%)** | | | |
| Secondary school | 5 (7) | 2 (3) | 7 (5) |
| TAFE or trade certificate (including diploma) | 10 (13) | 12 (16) | 22 (15) |
| Bachelor-level university degree | 35 (47) | 36 (48) | 71 (47) |
| Postgraduate university qualification | 25 (33) | 25 (33) | 50 (33) |
| **Other parent's education—n (%)** | | | |
| Secondary school | 5 (7) | 8 (11) | 13 (9) |
| TAFE or trade certificate (including diploma) | 24 (32) | 19 (25) | 43 (29) |
| Bachelor-level university degree | 26 (35) | 25 (33) | 51 (34) |
| Postgraduate university qualification | 20 (27) | 23 (31) | 43 (29) |
| **Maternal country of birth—n (%)** | | | |
| Australia | 51 (68) | 55 (73) | 106 (71) |
| United Kingdom | 6 (8) | 2 (3) | 8 (5) |
| Ireland | 3 (4) | 0 (0) | 3 (2) |
| Philippines | 1 (1) | 2 (3) | 3 (2) |
| Other | 8 (11) | 10 (13) | 18 (12) |
| Missing | 6 (8) | 6 (8) | 12 (8) |
| **Other parent's country of birth—n (%)** | | | |
| Australia | 53 (71) | 50 (67) | 103 (69) |
| United Kingdom | 3 (4) | 7 (9) | 10 (7) |
| New Zealand | 1 (1) | 4 (5) | 5 (3) |
| Ireland | 1 (1) | 3 (4) | 4 (3) |
| Other | 11 (15) | 4 (5) | 15 (10) |
| Missing | 6 (8) | 7 (9) | 13 (9) |
| **Delivery type—n (%)** | | | |
| Vaginal delivery | 25 (33) | 29 (39) | 54 (36) |
| Elective cesarean section | 21 (28) | 17 (23) | 38 (25) |
| Emergency cesarean section | 12 (16) | 18 (24) | 30 (20) |
| Forceps/vacuum-assisted delivery | 17 (23) | 11 (15) | 28 (19) |
| **Intrapartum antibiotics—n (%)** | | | |
| No | 48 (64) | 47 (63) | 95 (63) |
| Yes | 27 (36) | 27 (36) | 54 (36) |
| Missing | 0 (0) | 1 (1) | 1 (1) |

*(Continued)*

**Table 1.** (*Continued*)

| | First dose of aP combination vaccine (*n* = 75) | First dose of wP combination vaccine (*n* = 75) | Overall (*n* = 150) |
|---|---|---|---|
| **Neonatal antibiotics—n (%)** | | | |
| No | 66 (88) | 71 (95) | 137 (91) |
| Yes | 9 (12) | 4 (5) | 13 (9) |
| **Maternal gravidity—n (%)** | | | |
| 1 | 43 (57) | 31 (41) | 74 (49) |
| 2 | 18 (24) | 19 (25) | 37 (25) |
| 3 | 9 (12) | 14 (19) | 23 (15) |
| 4 | 3 (4) | 6 (8) | 9 (6) |
| 5+ | 2 (3) | 5 (7) | 7 (5) |
| **Maternal parity—n (%)** | | | |
| 1 | 53 (71) | 46 (61) | 99 (66) |
| 2 | 14 (19) | 22 (29) | 36 (24) |
| 3 | 7 (9) | 4 (5) | 11 (7) |
| 4 | 1 (1) | 3 (4) | 4 (3) |
| **Maternal dTpa in the preceding pregnancy—n (%)** | | | |
| 5c-dTpa | 21 (28) | 20 (27) | 41 (27) |
| 3c-dTpa | 43 (57) | 39 (52) | 82 (55) |
| Unknown | 11 (15) | 13 (17) | 24 (16) |
| None | 0 (0) | 3 (4) | 3 (2) |
| **Maternal dTpa last 5 years, excluding in the preceding pregnancy—n (%)** | | | |
| 5c-dTpa | 7 (9) | 6 (8) | 13 (9) |
| 3c-dTpa | 6 (8) | 9 (12) | 15 (10) |
| 3c-dTpa-IPV | 1 (1) | 0 (0) | 1 (1) |
| Unknown | 22 (29) | 25 (33) | 47 (31) |
| None | 39 (52) | 35 (47) | 74 (49) |
| **Maternal seasonal influenza vaccination during pregnancy—n (%)** | | | |
| No | 14 (19) | 10 (13) | 24 (16) |
| Yes | 61 (81) | 65 (87) | 126 (84) |
| **Mode of feeding—n (%)** | | | |
| Exclusively breastfed | 50 (67) | 52 (69) | 102 (68) |
| Exclusively formula-fed | 6 (8) | 4 (5) | 10 (7) |
| Both breastfed and formula-fed | 18 (24) | 19 (25) | 37 (25) |
| Both breastfed and started on solids | 1 (1) | 0 (0) | 1 (1) |
| **Cat ownership—n (%)** | | | |
| No | 59 (79) | 63 (84) | 122 (81) |
| Inside | 5 (7) | 3 (4) | 8 (5) |
| Inside and outside | 11 (15) | 9 (12) | 20 (13) |
| **Dog ownership—n (%)** | | | |
| No | 35 (47) | 39 (52) | 74 (49) |
| Inside | 14 (19) | 10 (13) | 24 (16) |
| Outside | 4 (5) | 3 (4) | 7 (5) |
| Inside and outside | 22 (29) | 23 (31) | 45 (30) |
| **Child attends day care—n (%)** | | | |
| No | 75 (100) | 75 (100) | 150 (100) |
| **Number of siblings—n (%)** | | | |

(*Continued*)

**Table 1.** (*Continued*)

| | First dose of aP combination vaccine (*n* = 75) | First dose of wP combination vaccine (*n* = 75) | Overall (*n* = 150) |
|---|---|---|---|
| 0 | 51 (68) | 41 (55) | 92 (61) |
| 1 | 14 (19) | 22 (29) | 36 (24) |
| 2 | 9 (12) | 5 (7) | 14 (9) |
| 3+ | 1 (1) | 4 (5) | 5 (3) |
| Missing | 0 (0) | 3 (4) | 3 (2) |
| **Apgar score, 1 minute—n (%)** | | | |
| <8 | 9 (12) | 9 (12) | 18 (12) |
| 8 | 5 (7) | 5 (7) | 10 (7) |
| 9 | 59 (79) | 61 (81) | 120 (80) |
| 10 | 1 (1) | 0 (0) | 1 (1) |
| Missing | 1 (1) | 0 (0) | 1 (1) |
| **Apgar score, 5 minutes—n (%)** | | | |
| <8 | 0 (0) | 3 (4) | 3 (2) |
| 8 | 2 (3) | 3 (4) | 5 (3) |
| 9 | 67 (89) | 64 (85) | 131 (87) |
| 10 | 5 (7) | 5 (7) | 10 (7) |
| Missing | 1 (1) | 0 (0) | 1 (1) |
| **Maternal ethnicity—n (%)[1]** | | | |
| European Caucasian | 68 (45) | 63 (42) | 131 (44) |
| Indian subcontinent | 7 (5) | 11 (7) | 18 (6) |
| Asian | 5 (3) | 9 (6) | 14 (5) |
| South American | 1 (1) | 2 (1) | 3 (1) |
| **Other parent's ethnicity—n (%)[1]** | | | |
| European Caucasian | 71 (47) | 72 (48) | 143 (48) |
| Indian subcontinent | 6 (4) | 4 (3) | 10 (3) |
| Asian | 3 (2) | 4 (3) | 7 (2) |
| South American | 0 (0) | 1 (1) | 1 (0) |
| Black African | 0 (0) | 1 (1) | 1 (0) |
| Indigenous Australia | 1 (1) | 0 (0) | 1 (0) |
| **Infant's ethnicity—n (%)[1]** | | | |
| European Caucasian | 72 (48) | 73 (49) | 145 (48) |
| Indian subcontinent | 10 (7) | 12 (8) | 22 (7) |
| Asian | 6 (4) | 11 (7) | 17 (6) |
| South American | 1 (1) | 2 (1) | 3 (1) |
| Black African | 0 (0) | 1 (1) | 1 (0) |
| Indigenous Australia | 1 (1) | 0 (0) | 1 (0) |
| **Birth measurements—Median (Q1–Q3)** | | | |
| Gestational age at delivery (weeks) | 39 (38–39) | 38 (38–39) | 38 (38–39) |
| Weight (g) | 3345 (3023–3601) | 3454 (3182–3765) | 3410 (3110–3664) |
| Weight missing—n (%) | 1 (1) | 0 (0) | 1 (1) |
| Length (cm) | 50 (49–51) | 51 (49–52) | 50 (49–51) |
| Length missing—n (%) | 1 (1) | 0 (0) | 1 (1) |
| Head circumference (cm) | 35 (34–36) | 35 (34–36) | 35 (34–36) |
| Head circumference missing—n (%) | 1 (1) | 0 (0) | 1 (1) |
| **Baseline measurements—Median (Q1–Q3)** | | | |

(*Continued*)

**Table 1.** (Continued)

| | First dose of aP combination vaccine (*n* = 75) | First dose of wP combination vaccine (*n* = 75) | Overall (*n* = 150) |
|---|---|---|---|
| Age (days) | 48 (45–50) | 47 (44–50) | 48 (45–50) |
| Weight (g) | 4830 (4308–5315) | 5025 (4515–5385) | 4888 (4361–5328) |
| Weight missing—n (%) | 0 (0) | 0 (0) | 0 (0) |
| Length (cm) | 56 (55–58) | 56 (55–58) | 56 (55–58) |
| Length missing—n (%) | 0 (0) | 0 (0) | 0 (0) |
| Head circumference (cm) | 38 (38–39) | 38 (38–40) | 38 (38–39) |
| Head circumference missing—n (%) | 1 (1) | 2 (3) | 3 (2) |

[1]Multiple ethnicities may apply, so percentages may not sum to 100.

aP: acellular pertussis vaccine. wP: whole-cell pertussis vaccine as a first dose. TAFE: technical and further education. 5c-dTpa: 5-component diphtheria-tetanus-acellular pertussis combination vaccine (reduced antigen formulation; includes pertussis toxoid, filamentous haemagglutinin, pertactin, and fimbriae types 2 and 3); 3c-dTpa: 3-component diphtheria-tetanus-acellular pertussis combination vaccine (reduced antigen formulation; includes pertussis toxoid, filamentous haemagglutinin, and pertactin); IPV: inactivated poliovirus vaccine.

IgE ≥ 0.01 and the difference between the mixed schedule group and the aP-only group for TT, egg white, and whole egg-IgE are presented in Table 3 (summaries for IgE ≥ 0.1 are included in Section D.4 in S1 Text). Despite exclusions, the results were consistent in the PP analysis set (Sections D.6–D.9 in S1 Text).

## Reactogenicity

The distribution of daily grades for solicited local and systemic adverse reactions following each vaccine occasion are presented in Figs 5–7 and in Section E in S1 Text. After the 6-week doses, 74 [99%] of 75 diary cards were returned in the mixed schedule group and 72 [96%] of 75 in the aP-only schedule group. Paracetamol use after the 6-week vaccine doses on more than one occasion was recorded among 59 [79%] of 75 wP and 52 [69%] of 75 aP recipients.

Overall, irritability was the most common solicited systemic reaction after a first dose of wP (65 [88%] of 74) versus aP (59 [82%] of 72). In addition, a first dose of wP was more frequently followed by any mild-to-moderate solicited local and systemic adverse reactions than aP (Sections E.1–E.2 in S1 Text).

Self-limited grade 3 (severe) erythema and swelling were reported at the pertussis vaccine (1 [1%] of 74) and 13vPCV (1 [1%] of 74) injection sites after a 6-week wP dose; none were reported after the 6-week aP dose (Fig 5 and Section E.2 in S1 Text). No reports of severe erythema, swelling, or induration at the pertussis vaccine or 13vPCV injection sites were recorded in either study group after the 4-month or 6-month vaccine doses. Severe pain (Fig 6 and Section E.2.4 in S1 Text) was recorded at the pertussis vaccine injection site after a 6-week wP dose (6 [8%] of 74), but not after a 6-week aP dose (0 of 72).

Fever ≥38.0˚C was reported in 0/73 infants vaccinated with wP and 1 (1%) of 70 infants vaccinated with aP at 6 weeks old, in 3 (4%) of 70 infants in the mixed schedule group and in 4 (6%) of 72 infants in the aP-only schedule group at 4 months old, and in 1 (1%) of 68 infants in each study group at 6 months old (Section E.1.7 in S1 Text).

The proportion of infants who experienced one or more severe (grade 3) solicited systemic adverse reactions was higher among those who received wP (14 [19%] of 74) than aP (8 [11%] of 72) at 6 weeks old, but similar after the 4-month (mixed schedule 9 [13%] of 71 versus aP-only schedule 12 [16%] of 73) and 6-month aP doses (mixed schedule 5 [7%] of 69 versus aP-only

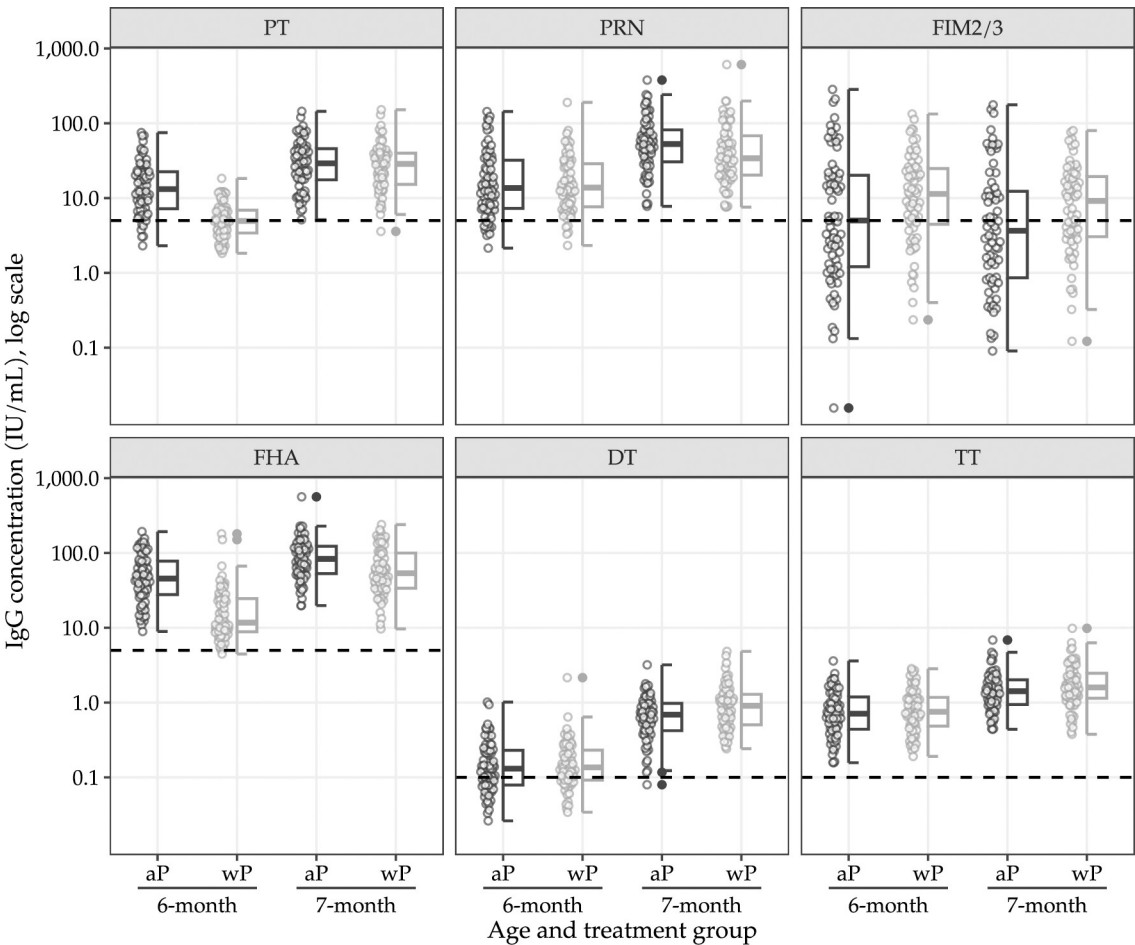

**Fig 2. IgG concentrations by antigen type, age, and assigned treatment.** Horizontal dotted line indicates antigen-specific seropositive threshold. aP: acellular pertussis vaccine. wP: whole-cell pertussis vaccine as a first dose. PT, pertussis toxin; PRN, pertactin, FIM2/3, fimbriae 2/3; FHA, filamentous hemagglutinin; DT, diphtheria toxin; TT, tetanus toxoid.

schedule 5 [7%] of 68). Severe irritability was reported in 11 [15%] of 74 wP versus 7 [10%] of 72 aP recipients at 6 weeks old, in 8 [11%] of 71 in the mixed schedule group and 8 [11%] of 73 in the aP-only schedule group at 4 months old, and in 5 [7%] of 69 in the mixed schedule group versus 4 [6%] of 68 in the aP-only schedule group at 6 months old (Fig 7 and Section E.1 in S1 Text). Four infants assigned to the aP-only schedule (4 [6%] of 72) and 5 infants assigned to the mixed schedule (5 [7%] of 74) experienced grade 3 solicited systemic adverse reactions on 2 occasions. No severe recurrent solicited adverse reactions were reported in the remaining infants.

## Safety

There were 7 SAEs among 5 participants within the first 6 months of follow-up; on blinded assessment, none were assessed to be related to the study vaccines. No hypotonic hyporesponsive episodes or breakthrough pertussis infections occurred in either group.

## Parental acceptability

On Day 6 postvaccination at 6 weeks old, 71 (97%) of 73 parents of wP recipients and 69 (96%) of 72 parents of aP recipients either agreed or strongly agreed that they would be happy for their child to receive the same vaccine combination (Section E.3 in S1 Text).

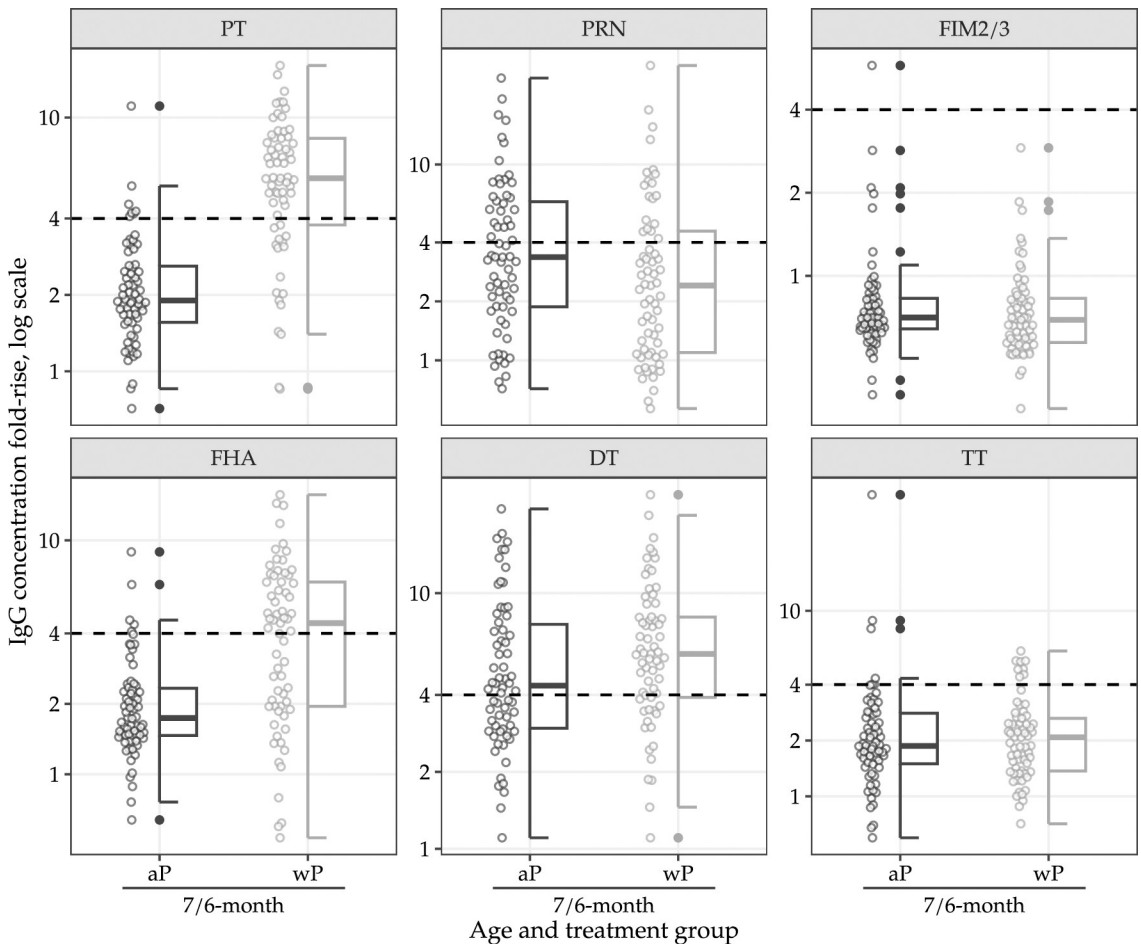

**Fig 3. IgG fold-rise by type, age, and assigned treatment.** Horizontal dotted line indicates 4-fold rise. aP: acellular pertussis vaccine. wP: whole-cell pertussis vaccine as a first dose. PT, pertussis toxin; PRN, pertactin, FIM2/3, fimbriae 2/3; FHA, filamentous hemagglutinin; DT, diphtheria toxin; TT, tetanus toxoid.

## Discussion

We report the results of a randomised comparison of the immunogenicity, reactogenicity, and IgE-mediated immune responses to a mixed wP/aP primary vaccine schedule versus the standard aP-only schedule. Mixed priming (wP/aP/aP) using the WHO-prequalified 5-in-1 wP vaccine (Pentabio, PT Bio Farma, Indonesia) and 3c-aP vaccine formulation containing PT, FHA, and PRN pertussis antigens (Infanrix hexa, GlaxoSmithKline, Australia) was noninferior to the homologous aP-only schedule (aP/aP/aP) with respect to anti-PT antibody titres at 6 and 7 months old. While at 7 months old almost all (99%) serum responses to DT and TT met their specified seroprotective thresholds and the IgG responses to FHA and PRN met the specified seropositivity cutoff in the 2 treatment groups, PT and FHA-IgG concentrations immediately before the 6-month aP dose were lower in the mixed schedule group compared to the aP-only group. Given the lower content of PT and FHA in wP versus aP-based vaccines, this finding was not unexpected; by 1 month after the 6-month aP dose, PT concentrations were similar, but concentrations of FHA remained lower in the mixed schedule compared to the aP-only schedule group. IgG GMCs for FIM2/3 were low at 6 and 7 months in both groups. Infanrix hexa has no FIM2/3 antigenic content; in the absence of natural infection, a progressive

**Table 2. IgG GMR posterior summaries (mixed schedule group/acellular pertussis vaccine-only group) and noninferiority probability, for diphtheria, tetanus, and pertussis antigens.**

| | Mean ± standard deviation | Median | 95% Credible interval | Probability (>1) | Probability (>2/3) |
|---|---|---|---|---|---|
| **Diphtheria toxoid** | | | | | |
| 6-month | 1·11 ± 0·14 | 1·10 | (0·85, 1·42) | 0·77 | >0·99 |
| 7-month | 1·36 ± 0·16 | 1·35 | (1·07, 1·70) | 0·99 | >0·99 |
| 7/6-month geometric mean fold rise | 1·24 ± 0·14 | 1·23 | (0·97, 1·55) | 0·96 | — |
| **Filamentous haemagglutinin** | | | | | |
| 6-month | 0·36 ± 0·05 | 0·36 | (0·28, 0·46) | 0·00 | 0·00 |
| 7-month | 0·70 ± 0·08 | 0·69 | (0·56, 0·87) | 0·00 | 0·67 |
| 7/6-month geometric mean fold rise | 1·97 ± 0·24 | 1·96 | (1·55, 2·47) | > 0·99 | — |
| **Fimbriae 2/3** | | | | | |
| 6-month | 1·96 ± 0·59 | 1·88 | (1·04, 3·34) | 0·98 | >0·99 |
| 7-month | 1·85 ± 0·53 | 1·78 | (1·01, 3·09) | 0·98 | >0·99 |
| 7/6-month geometric mean fold rise | 0·95 ± 0·07 | 0·95 | (0·82, 1·09) | 0·23 | — |
| **Pertactin** | | | | | |
| 6-month | 1·03 ± 0·17 | 1·01 | (0·74, 1·40) | 0·53 | >0·99 |
| 7-month | 0·80 ± 0·12 | 0·79 | (0·59, 1·07) | 0·06 | 0·88 |
| 7/6-month geometric mean fold rise | 0·79 ± 0·13 | 0·78 | (0·57, 1·06) | 0·06 | — |
| **Pertussis toxin** | | | | | |
| 6-month | 0·37 ± 0·04 | 0·37 | (0·30, 0·46) | 0·00 | 0·00 |
| 7-month | 0·99 ± 0·13 | 0·98 | (0·77, 1·26) | 0·43 | >0·99 |
| 7/6-month geometric mean fold rise | 2·64 ± 0·27 | 2·63 | (2·15, 3·21) | > 0·99 | — |
| **Tetanus toxoid** | | | | | |
| 6-month | 1·16 ± 0·14 | 1·15 | (0·92, 1·45) | 0·89 | > 0·99 |
| 7-month | 1·16 ± 0·12 | 1·15 | (0·93, 1·42) | 0·91 | > 0·99 |
| 7/6-month geometric mean fold rise | 1·01 ± 0·10 | 1·00 | (0·83, 1·21) | 0·50 | — |

GMR, geometric mean ratio.

All models adjusted for sex, birth order, breastfeeding status, delivery method, family history of atopic disease, and parental income as baseline covariates.

decline in anti-FIM2/3 antibodies, which are either maternally derived or induced by a single dose of wP is expected.

Previous studies have found concentrations of antibodies to aP vaccine antigens were either similar or lower in infants primed with wP than in those primed with aP [30]. In addition, it has been speculated that maternal vaccination might cause greater interference of the immunogenicity of wP than aP vaccines in early infancy. In our trial, almost all mothers had received aP in pregnancy, so their infants are likely to have had high levels of maternally derived antibodies targeting aP vaccine antigens at the time of receipt of the 6-week vaccine doses.

A dose of wP at approximately 6 weeks old was more reactogenic than a dose of aP, with a higher frequency of solicited local and systemic adverse reactions, which were mostly mild-to-moderate in severity. While after the 6-week doses severe local adverse reactions were only described at the wP and 13vPCV injection sites, fever $\geq38.0\,^{\circ}C$ was only described in the aP-vaccine study group. The proportion of parents of wP-vaccinated infants who reported acceptability of the vaccination was very high and similar to that of parents of aP-vaccinated infants. The young age of administration of first dose of the vaccines and the routine use of prophylactic paracetamol may have attenuated the intensity of solicited reactions and possibly enhanced parental acceptance of wP [31]. The reactogenicity data were presented quarterly to an independent safety and monitoring committee, which supported continued enrolment.

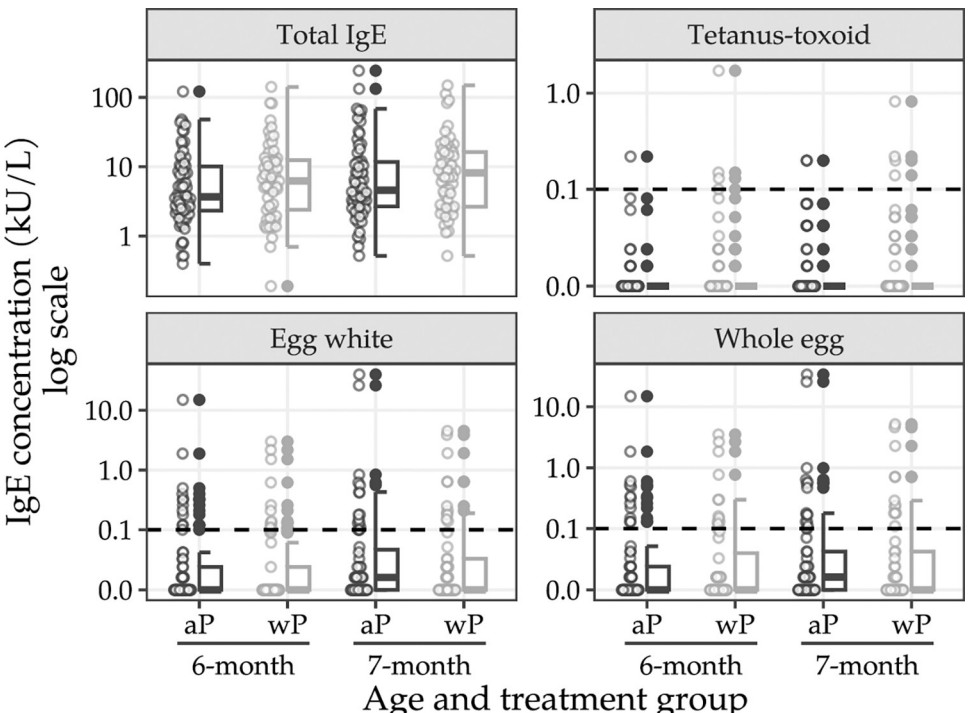

**Fig 4. IgE concentrations by allergen type, age, and assigned treatment.** Horizontal dotted line indicates antigen-specific seropositive threshold. IgE concentrations by allergen type, age, and assigned schedule.

**Table 3. Posterior summaries for IgE quantitation probability and difference for tetanus toxoid and hen's egg by age and assigned treatment.**

| | Standardised probability (IgE ≥ 0.01), median (95% credible interval) | | | | Odds ratios | |
|---|---|---|---|---|---|---|
| | First dose of aP combination vaccine (aP) | First dose of wP combination vaccine (wP) | wP-aP | Probability (wP-aP <0) | Conditional | Marginal |
| **Tetanus toxoid IgE** | | | | | | |
| 6-month | 0·16 (0·08, 0·27) | 0·21 (0·11, 0·33) | 0·04 (−0·05, 0·15) | 0·20 | 2·61 (0·27, 35·40) | 1·33 (0·67, 3·11) |
| 7-month | 0·22 (0·12, 0·34) | 0·22 (0·12, 0·34) | 0·00 (−0·12, 0·12) | 0·52 | 0·95 (0·06, 13·91) | 0·98 (0·43, 2·26) |
| **Egg white IgE** | | | | | | |
| 6-month | 0·35 (0·24, 0·48) | 0·28 (0·17, 0·41) | −0·07 (−0·20, 0·05) | 0·86 | 0·36 (0·04, 2·35) | 0·72 (0·35, 1·31) |
| 7-month | 0·49 (0·37, 0·62) | 0·46 (0·33, 0·59) | −0·03 (−0·19, 0·11) | 0·68 | 0·63 (0·07, 5·67) | 0·86 (0·43, 1·67) |
| **Whole egg IgE** | | | | | | |
| 6-month | 0·41 (0·30, 0·54) | 0·41 (0·28, 0·53) | −0·01 (−0·13, 0·12) | 0·55 | 0·88 (0·11, 6·97) | 0·96 (0·55, 1·72) |
| 7-month | 0·49 (0·37, 0·62) | 0·41 (0·28, 0·53) | −0·06 (−0·21, 0·08) | 0·81 | 0·36 (0·03, 3·55) | 0·75 (0·38, 1·43) |

aP, acellular pertussis vaccine; wP, whole-cell pertussis vaccine. First dose of aP combination vaccine (aP): aP-only schedule. First dose of wP combination vaccine (wP): mixed schedule. All models adjusted for sex, birth order, breastfeeding status, delivery method, family history of atopic disease, and parental income as baseline covariates.

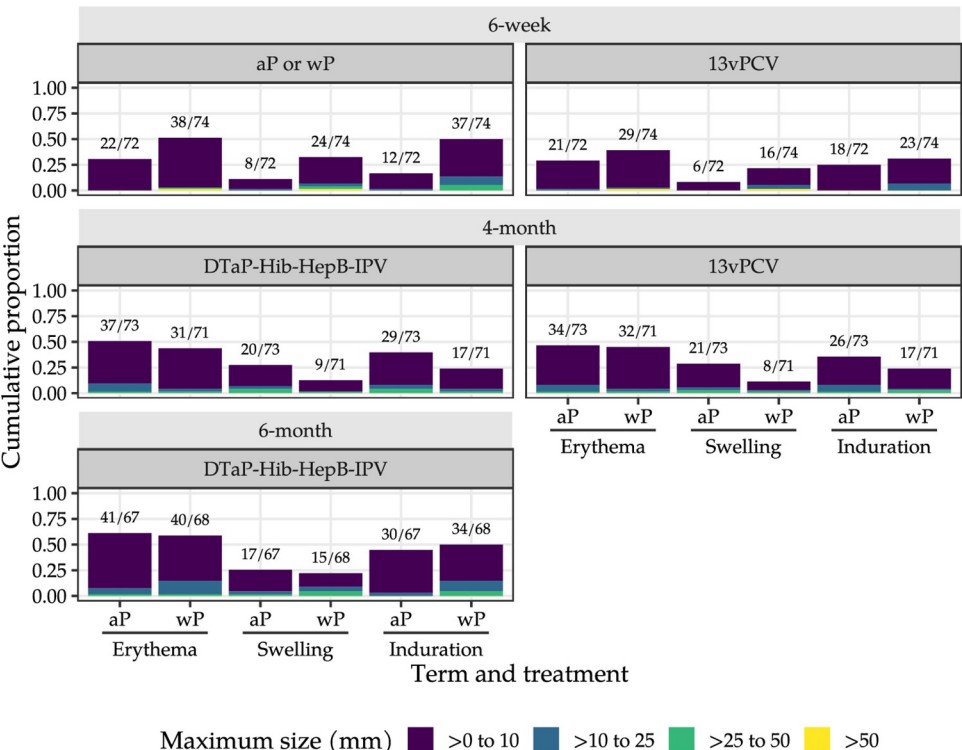

**Fig 5. Maximum injection site reaction size in the 7 days after the 6-week, 4-month, and 6-month vaccine doses.**
aP, acellular pertussis vaccine; wP, whole-cell pertussis vaccine; 13vPCV: 13-valent pneumococcal conjugate vaccine; DTaP-Hib-HepB-IPV, hexavalent aP vaccine (diphtheria, tetanus, aP, *Haemophilus influenzae* type b, hepatitis B, and inactivated poliovirus types 1, 2, and 3 vaccine). Above each bar (n/N), we indicate the number of infants reporting any injection site reaction > 0 mm (numerator; n) versus the total number of infants reporting on injection site reactions ≥ 0 mm (denominator; N).

A previous RCT suggested that therapeutic paracetamol administered within 48 hours post-DTwP vaccination does not interfere with IgG immune responses elicited by the vaccine antigens in children receiving a homologous 2-, 4-, 6-, and 18-month schedule [32]. More recently, a trial reported that prophylaxis with oral paracetamol had no effect on immunogenicity of a combination aP vaccine, coadministered with 7-valent-PCV and a recombinant multicomponent meningococcal B vaccine at 2, 3, and 4 months old [33]. By contrast, in infants receiving a combination aP vaccine coadministered with the 10-valent pneumococcal nontypeable *Haemophilus influenzae* protein D-conjugate vaccine (PhiD-CV) at 3, 4, and 5 months old, prophylactic rectal paracetamol was associated with reduced IgG responses to aP antigens and pneumococcal vaccine polysaccharides [34]. While the clinical significance of these findings is uncertain, a follow-up study suggested that paracetamol prophylaxis had no impact on the induction of PhiD-CV-serotype-specific immunological memory or pneumococcal nasopharyngeal carriage as measured at 4 years old [35].

Owing to evidence of improved long-term pertussis protection among wP vaccine recipients, WHO recommends that countries using wP should only changeover to less reactogenic aP-only schedules where it is financially and programmatically feasible to provide frequent aP boosters, including in pregnancy [36]. For countries that have already transitioned to an aP-only schedules, mixed schedules may provide better long-lived protection against pertussis but confirmation of this would require the follow-up of many thousands of infants over many years.

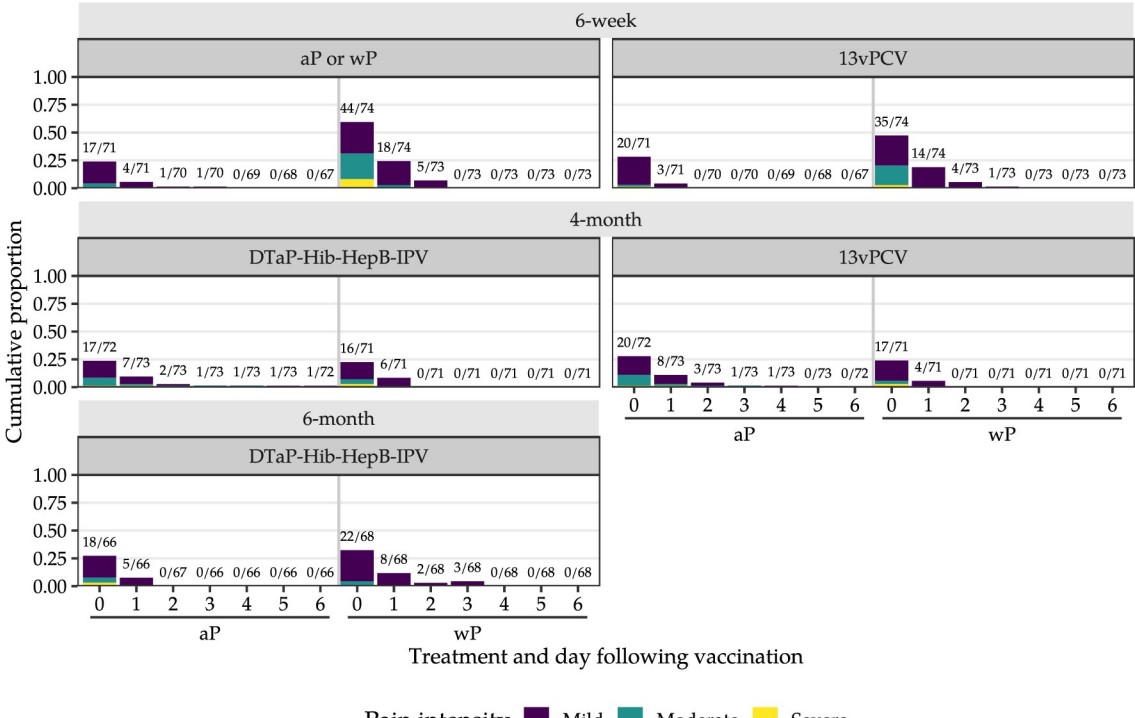

**Fig 6. Injection site pain in the 7 days after the 6-week, 4-month, and 6-month vaccine doses.** aP, acellular pertussis vaccine; wP, whole-cell pertussis vaccine; 13vPCV: 13-valent pneumococcal conjugate vaccine; DTaP-Hib-HepB-IPV, hexavalent aP vaccine (diphtheria, tetanus, aP, Haemophilus influenzae type b, hepatitis B, and inactivated poliovirus types 1, 2, and 3 vaccine); pain intensity, mild (minor reaction on touch or does not interfere with daily activities); moderate (cries/protests on touch or interferes with daily activities); severe (cries when limb is moved/spontaneously painful or prevents daily activities). Above each bar (n/N), we indicate the number of infants reporting any pain at the injection site (i.e., mild, moderate, or severe pain; numerator; n) versus the total number of infants reporting on pain at the injection site (none, mild, moderate, or severe pain; denominator; N).

Examining the immune responses driven by wP vaccine formulations is complex. The exact concentrations of pertussis antigens within wP varies across formulations. Previous clinical trials reported differences in immune responses between wP- and aP-based vaccines administered in homologous priming schedules and across various specific formulations of the 2 vaccine types. Antibody responses to aP-specific antigens are generally higher among aP-vaccinated infants than wP-vaccinated infants [37,38]. However, greater vaccine effectiveness has been documented among school-aged children and adolescents who had received wP versus aP as their first dose [4]. Vaccine responses have been traditionally assessed by measuring antibody responses only. Pertussis-specific T cell memory responses may be important in long-term protection induced by pertussis vaccines [8]. Future studies will compare the patterns of CD4+ T cell polarisation in response to pertussis vaccine antigens in a subcohort of infants enrolled in stage one.

Observational studies suggest that laboratory-confirmed pertussis may be less likely among wP-primed versus aP-primed children, and among those receiving mixed schedules in which the first dose was wP versus aP. A case–control study found that pertussis disease was less common among children receiving a mixed schedule of wP/3c-aP versus 3c-aP only [39]. No evidence was found of a difference in the risk of pertussis among those vaccinated with a mixed wP/5c-aP schedule versus those vaccinated with a with a 5c-aP formulation, although long-term differences cannot be excluded [39]. Another case–control study found that compared to a 5c-aP-only primary series, vaccination with one or more primary doses of a low-efficacy wP

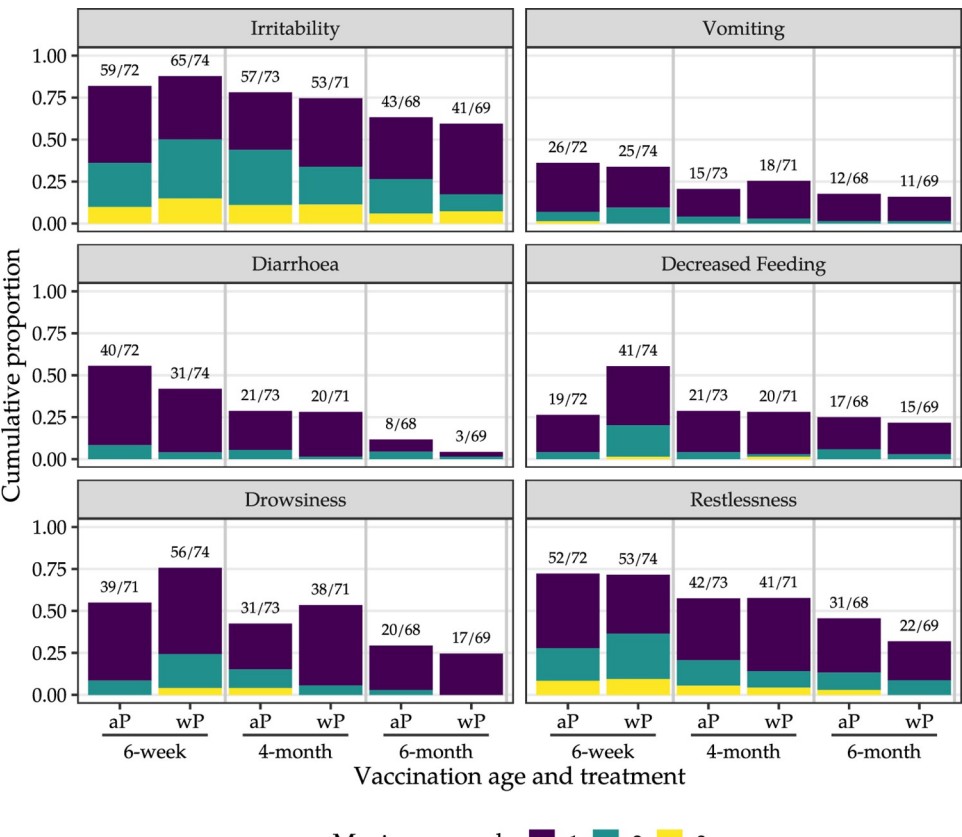

**Fig 7. Highest intensity grade for systemic reactions after the 6-week, 4-month, and 6-month vaccine doses.** aP, acellular pertussis vaccine; wP, whole-cell pertussis vaccine; grade 1: easily tolerated; grade 2, interferes with normal everyday activities; grade 3, prevents normal everyday activities or requires significant medical intervention. Above each bar (n/N), we indicate the number of infants reporting systemic reactions grade 1 and above (numerator; n) versus the total number of infants reporting on systemic reactions (none, grade 1, grade 2, or grade 3; denominator; N).

vaccine formulation was associated with a lower risk of pertussis disease more than a decade after priming [40]. None of the cited case–control analyses provided further details regarding the nature of the first pertussis vaccine dose in the mixed schedules examined.

WHO-prequalified 5-in-1 wP vaccine formulations have been successfully introduced across 77 lower-income countries supported by Gavi. While earlier concerns about safety led to the discontinuation of wP in Australia and other high-income countries, a 2021 meta-analysis of 15 RCTs (38,072 infants) found that any risk difference for SAEs in infants primed with wP versus aP is likely to be small, ranging from 3 fewer to 2 more events per 1,000 children [41]. Similar to our trial, the meta-analysis defined SAEs as event that resulted in death, were life-threatening, required hospitalisation or prolongation of existing hospitalisation, or resulted in persistent or significant disability or incapacity. Our data are therefore consistent with the excellent safety record of wP previously highlighted by WHO and support the ongoing use of wP.

We have previously observed that Australian children with IgE-mediated food allergy were less likely than contemporaneous controls to have received one or more doses of wP in infancy [5]. We hypothesised that compared to a standard aP-only schedule, a mixed vaccine schedule comprising an initial dose of wP could help promote the normal transition from a Th$_2$-skewed

to a balanced $Th_1/Th_{17}/Th_2$ immunophenotype in early infancy and thereby protect against IgE-mediated food allergy. Having previously noted that wP-primed infants were less likely than aP-primed infants to develop boosted vaccine-associated IgE responses [10], we sought to confirm this observation by testing for total, TT, and egg-specific IgE at 6 and 7 months. While the production of specific IgE against DTaP antigens is known to only occur in a subset of vaccinated infants [11], children [10], or adults [12], the low TT-IgE concentrations observed in our study are not easily explained. The detection of total and specific IgE were conducted using an autoanalyser IgE-based assay, and, therefore, interference by anti-allergen-specific IgG is unlikely [26]. However, to our knowledge, except for analyses carried out in Australia [26], Belgium [9], and the United States [42], TT-specific IgE responses have been measured using RAST or other serological methods. Thus, apart from the first group of cited studies, it is not possible to directly compare our IgE results with prior cohorts.

To our knowledge, this study represents the first prospective randomised comparison of a novel mixed wP/aP versus aP-only schedule. We used a prequalified 5-in-1 combination vaccine formulation already widely adopted and available in Gavi-supported countries, and the immunogenicity of each schedule was assessed using a validated multiplex fluorescent bead-based immunoassay.

The limitations of this study include uncertain generalisability since the study population comprised a high proportion of infants of privately insured urban parents. Secondly, while our previous case–control study suggests that the potential allergy protective benefits of wP might be confined to IgE-mediated peanut or tree nuts allergy [5], stage one did not examine IgE responses to these antigens. Apart from the difference between regimens in wP content, there were also differences in the content of IPV (absent in Pentabio PT Bio Farma), aluminium (lower in Pentabio PT Bio Farma), and adventitial antigens [16]. We can therefore not be certain whether any observed differences between regimens can be attributed to the difference in the wP content or to these other factors. In addition, we were not powered to compare schedules for important clinical outcomes, including pertussis disease. The mixed schedule was well accepted by the cohort of parents who consented to participate after being informed of the known reactogenicity profile of wP; it is unclear whether the high acceptance of wP observed is generalisable to all parents. We note that many parents were motivated to participate because of a desire to prevent IgE-mediated food allergies in their infants. Further evidence is required to understand the population-level acceptability of this approach beyond the study population. In conclusion, our findings give support to the acceptable immunogenicity and reactogenicity of the mixed primary schedule. These findings are relevant to countries where both wP and aP vaccines are used and support the further evaluation of the clinical effects of mixed schedules. We failed to find confirming immunological evidence of an attenuating effect of the mixed versus the aP-only schedule on TT or egg-related IgE antibodies; owing to the unclear relationship between these biomarkers and the subsequent development of IgE-mediated food allergy, a clinically important effect cannot be excluded. In stage two, the effect of the mixed versus standard schedule on the development of IgE-mediated food allergy by 12 months will be assessed in up to 3,000 infants.

## Supporting information

**S1 Text. Supporting information.**
(PDF)

**S1 CONSORT Checklist. CONSORT 2010 checklist of information to include when reporting a randomised trial.**
(DOC)

## Acknowledgments

We acknowledge Aboriginal and Torres Strait Islander People as the Traditional Custodians of the land and waters of Australia. We also acknowledge the Nyoongar Wadjuk, Yawuru, Kariyarra, and Kaurna Elders, their people and their land upon which the Telethon Kids Institute is located, and where stage one of the OPTIMUM trial was conducted. We seek their wisdom in our work to improve the health and development of all children. We thank the parents and infants who took part in stage one of this trial for their commitment and continuous support. We acknowledge all members of the Vaccine Trials Group at the Telethon Kids Institute, who were involved in the conduct of stage one of the OPTIMUM study. We would also like to express our gratitude to Kylie Rogers (data manager) and Dr Andrew McLean-Tooke (Head– PathWest Immunology QEII campus, Western Australia, Australia) for their valuable contributions, and to the independent data safety monitoring committee for their detailed oversight. We thank Catherine Hughes (AM), consumer representative in the OPTIMUM study Steering Committee Group for her valuable input in the design and conduct of this study.

## Author Contributions

**Conceptualization:** Gladymar Pérez Chacón, Julie A. Marsh, Kirsten P. Perrett, Dianne E. Campbell, Claire S. Waddington, Nigel Curtis, Peter B. McIntyre, Patrick G. Holt, Peter C. Richmond, Tom Snelling.

**Data curation:** Gladymar Pérez Chacón, James Totterdell.

**Formal analysis:** James Totterdell, Sonia McAlister.

**Funding acquisition:** Julie A. Marsh, Kirsten P. Perrett, Dianne E. Campbell, Nicholas Wood, Michael Gold, Peter C. Richmond, Tom Snelling.

**Investigation:** Gladymar Pérez Chacón, Marie J. Estcourt, James Totterdell, Michael O' Sullivan, Sonia McAlister.

**Methodology:** Marie J. Estcourt, James Totterdell, Julie A. Marsh, Kirsten P. Perrett, Dianne E. Campbell, Claire S. Waddington, Sonia McAlister, Nigel Curtis, Peter C. Richmond, Tom Snelling.

**Project administration:** Marie J. Estcourt.

**Software:** James Totterdell.

**Supervision:** Marie J. Estcourt, Michael O' Sullivan, Peter C. Richmond, Tom Snelling.

**Validation:** James Totterdell, Sonia McAlister, Mark Jones.

**Visualization:** James Totterdell.

**Writing – original draft:** Gladymar Pérez Chacón.

**Writing – review & editing:** Gladymar Pérez Chacón, Marie J. Estcourt, James Totterdell, Julie A. Marsh, Kirsten P. Perrett, Dianne E. Campbell, Nicholas Wood, Michael Gold, Claire S. Waddington, Michael O' Sullivan, Sonia McAlister, Nigel Curtis, Mark Jones, Patrick G. Holt, Peter C. Richmond, Tom Snelling.

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
