## [Editor Report · Decision Letter 0]

22 Jan 2024

Dear Dr Pérez Chacón, 

Thank you for submitting your manuscript entitled "Immunogenicity, reactogenicity, and IgE-mediated immune responses of a mixed whole-cell and acellular pertussis vaccine schedule in Australian infants: a randomised, double-blind, non-inferiority trial" for consideration by PLOS Medicine.

Your manuscript has now been evaluated by the PLOS Medicine editorial staff and I am writing to let you know that we would like to send your submission out for external peer review.

Please re-submit your manuscript within two working days, i.e. by Jan 24 2024 11:59PM.

Feel free to email me at pdodd@plos.org or the team at plosmedicine@plos.org if you have any queries relating to your submission.

Kind regards,

Pippa

Philippa Dodd, MBBS MRCP PhD

PLOS Medicine

---

## [Decision Letter · Decision Letter 1]

21 Feb 2024

Dear Dr. Pérez Chacón,

Many thanks for submitting your manuscript to PLOS Medicine. The paper has been reviewed by three subject experts and a statistician; their comments are included below and can also be accessed here: [LINK].

As you will see, the reviewers were generally supportive of the paper and the clarity of the reporting, although there was a consensus that inclusion of functional T-cell data would substantially increase the value of and interest in the paper. The Editors agree with this, and we strongly encourage you to add T-cell data if possible, although we acknowledge that this was not prespecified in the protocol (as such, this would need to be presented as a post-hoc/exploratory analysis). 

Based on the reviewers’ comments and discussion with an Academic Editor with relevant expertise, I’m pleased to invite you to revise the paper in response to the reviewers’ comments. We plan to send the revised paper back to the original reviewers, and of course we cannot provide any guarantees at this stage regarding publication. 

When you upload your revision, please include a point-by-point response* that addresses all of the reviewer and editorial points, indicating the changes made in the manuscript and either an excerpt of the revised text or the location (eg: page and line number) where each change can be found. Please submit a clean version of the paper as the main article file and a version with changes marked should as a marked-up manuscript. Please also check the guidelines for revised papers at http://journals.plos.org/plosmedicine/s/revising-your-manuscript for any that apply to your paper. 

We ask that you submit your revision by March 13th. However, if this deadline is not feasible, please contact me by email, and we can discuss a suitable alternative. 

Don’t hesitate to contact me directly with any questions (hvanepps@plos.org). If you reply directly to this message, please be sure to ‘Reply All’ so your message comes directly to my inbox. 

Kind regards, 

Heather

Heather Van Epps, PhD

Executive Editor

PLOS Medicine

hvanepps@plos.org

*Please note while forming your response, if your article is accepted, you may have the opportunity to make the peer review history publicly available. The record will include editor decision letters (with reviews) and your responses to reviewer comments. If eligible, we will contact you to opt in or out.

Reviewer comments:

Reviewer #1:

[Numbering added by editor]

In this study the authors report the safety and immunogenicity data from a RCT that investigates the effect of the primary dose of pertussis vaccination. The study is generally very well executed and addresses an interesting and important concept, i.e. that different pertussis vaccine types with known distinct effects on immunological imprinting against pertussis antigens may also have off-target effects, such as IgE-mediated food allergy. Even though no differences were observed with regards to the IgE measurements in the current smaller study, it will be very interesting to see what happens in the larger cohort. 

1. Whilst the study and reported endpoints are clearly described, please find below a number of comments/questions that the authors may wish to address:

2. The authors focus on the comparison between the acellular and the whole cell pertussis vaccine. However, given that the pentavalent whole cell vaccine does not contain IPV and the hexavalent acellular vaccine does, and that a standalone dose of IPV was not administered together with the pentavalent wP at 2m of age but rather at the age of 18m (i.e. after the immunological endpoints), the authors should discuss the potential role of IPV as it cannot be formally ruled out that it also plays a role. 

3. Based on the methods section I assume that the assay was run blinded and randomized. Were the samples individually paired on the same run/batch? How was QC done in the immunological assays? And what was the reproducibility of the assays? This is particularly relevant for the IgE measurements that seem less standardized (I may be wrong). 

4. It is not always very clearly described for each modeled immune response analysis in the main/supplementary data which parameters are included. 

5. The authors report higher solicited systemic AEs in the wP-group compared to the aP-group (lines 307-310) at 6w of age but not after the subsequent doses. Were AEs reported at the different time points observed in the same infants or were these different infants? 

6. Figures 2/3 in the main paper show the antibody responses at the 6 and 7m time points. It would be helpful if these figures would also show whether differences between groups/time points are statistically different or not. 

Reviewer #2:

[Numbering added by editor]

In this study, Gladymar Pérez Chacón and colleagues analyzed the immunogenicity and reactogenicity of a mixed pertussis vaccination schedule in a randomized double-blind clinical trial. The novelty of this study resides in the direct comparison of the effect of a whole-cell pertussis vaccine administered as the first priming vaccination compared to an acellular pertussis vaccine. The aim was to understand not only the implications of reactogenicity and immunogenicity but also the IgE-mediated immune responses and possible associations of acellular pertussis vaccines with the development of IgE-mediated food allergies. Although this trial is well designed and conducted, the results of this report only concern stage 1, which reports relatively fewer individuals and short time points and do not include stage 2, which encompasses a later time point. However, stage 2 addresses a much more important question and covers more individuals by a large margin. 

Nevertheless, I believe the manuscript is of interest and could be improved by addressing the suggestions below. 

General comments:

1. My primary concern is the fact that the findings on IgE reactivity are fairly inconclusive since they only pertain to stage 1 of the trial. The study would be much more interesting if data from stage 2 were combined with the present manuscript. Alternatively, a manuscript solely focused on reactogenicity and immunogenicity would still address a solid concern associated with the mixed vaccine schedule and would not leave the reader wondering if an effect of IgE could or could not be observed in the next chapter. 

2. Although the authors acknowledge that T cell responses are important and will be evaluated, no immunogenicity data is present in the current manuscript. I believe that this data combined with serology would be much more informative and interesting for the readership of this Journal.

Specific comments:

3. Abstract: Both findings and conclusions do not mention IgE responses, which would be expected since it is stated in the title. As per my suggestions above, I would remove entirely this topic from this manuscript since I don't think Stage 1 is equipped to answer what I think is one of the main and most important goals of this study. [Editors’ note: while we appreciate the opinion of the reviewer, the outcomes should be reported as prespecified in the study protocol.]

Introduction:

4. Line 76 - References are more than 10 years old. I am wondering if there are more compelling and recent evidences.

5. Line 83 - When discussing the impact on food allergy, it would be interesting also to note that similar studies showed no differences in childhood asthma (PMID: 37792889). Also, it would be interesting to point out that two other studies showed differential expression of IgE in aP- vs wP-primed cohorts for either pertussis or non-pertussis vaccine antigens (PMID: 31419322, PMID: 33690224).

6. Line 98 - The reviewer wonders what the rationale was for choosing 2 time points so close to each other.

Results:

7. Overall, the structure would benefit from more informative subheadings.

8. Line 229 - The supplementary document is overwhelming with too much information and little to no context besides a figure legend. Although I understand that these are important documentations, I wonder if all are really necessary for the readership of this journal and/or if another online repository of clinical trial data and a link wouldn't be more suitable. Also, Figure 2.1 is exactly the same as Figure 2 in the manuscript body. Figure 2.3 is interesting and should be mentioned since it could reflect the different kinetics of two different pertussis immunization schedules. It would also be very interesting to understand these dynamics at the 12-month or over time points, which would be a key point on durability of protection.

9. Line 239 - The authors should also comment on the lower seroconversion of Fim 2/3.

10. The statistical differences are difficult to interpret due to the complexity of the statistical methodology. I am curious if there is a simpler method available to summarize these differences. [Editors’ note: As above, the statistical analysis should follow the prespecified protocol/SAP.]

11. Also, why was IgE responses to pertussis vaccine antigens not performed?

Discussion:

12. Some study limitations are presented throughout the well-written discussion but could be all condensed in a unique paragraph. Can you also develop on how and what is going to be exactly evaluated in the assessment of T cell responses?

Reviewer #3: 

This manuscript reports the results of a clinical trial to assess the possibility that a single dose of a whole cell pertussis (wP) vaccine given before the acellular pertussis (aP) vaccine might prevent or reduce IgE responses to food allergens. However, the results revealed no benefit in terms of reduced IgE response to egg allergens and as expect the wP vaccine was associated with higher adverse events. 

Specific comments: 

1. The wP vaccine was discontinued in most developed countries in the 1990s. While the aP vaccine is clearly inferior to the wP vaccine, especially at preventing nasal infection and transmission of B. pertussis, it seems a retrograde step to go back to wP, even for a single dose in the primary immunization schedule. Alternatives to the current aP vaccine, including attenuated vaccines, aP vaccines with novel adjuvants and less toxic wP vaccines are in development and these would appear to be a better solution to the less-than-ideal current aP vaccine. While the parents of the children in this trial may have indicated acceptance of the side effects associated with the wP vaccine, this is unlikely to the general view from health care professionals, regulators and parents globally.

2. Previous studies have reported that aP vaccines do not boost IgE response to common allergens, despite promoting Th2 and IgE response to aP vaccine antigens. The current report references previous studies that have suggested that Australian children who received wP, rather than aP, vaccine as their first immunization had less IgE-mediated food allergy. So the question remains, did immunization with aP vaccine only promote IgE response to allergens (over that which might be developed irrespective of vaccination) or did the wP vaccine induce some "protective" effect against general food allergen-induced IgE and if so, how. If the wP vaccine is "protective", this boils down to giving children a reactogenic vaccine to reduce the incidence of food allergy. 

3. The underline hypothesis for this study seems to have been based on the premise that the Th2 and associated IgE response induced by the aP vaccine would be skewed more to Th1/Th17 with a primary dose of the wP vaccine and this would reduce/eliminate the IgE responses. However, the study has not assessed or reported any data on T cell responses from the clinical trial. This is a significant omission and weakens the impact of the study.

4. The presentation of the results, especially the narrative in the text of the results section is weak, with very limited commentary on the data, their meaning or significance. 

5. The authors use the term "seropositive/seroprotective threshold" when referring to IgG responses to B. pertussis antigens. There is no correlate of protective immunity against B. pertussis, so could they please explain what is a seroprotective concentration of antibody and against what antigen(s)?

6. The study has used a wP vaccine from PT Biofarma, Indonesia (and this is not mentioned until the discussion section). Detailed information and/or references to studies using this vaccine should be provided in the Methods section of the manuscript. 

Reviewer #4: Alex McConnachie, Statistical Review 

[Numbering added by editor]:

The paper by Pérez Chacón et al reports on the initial phase of the larger OPTIMUM trial, in which the immune responses and safety of two vaccination schedules are compared. This review considers the statistical aspects of the paper.

Overall, I found the paper to be nicely written, and the statistical methods and application seem very good. The data presentation is quite clear, with some very nice graphs. I do have a few comments, which are fairly minor. I shall describe these in the order in which I encountered them in the paper.

1. The primary analysis is by intention to treat. For a non-inferiority trial, some would argue that the primary analysis should be for a per-protocol population, though I think it is fine as long as both are reported. The PP results are somewhat hidden away in the supplements, and do not appear to be commented on in the paper. I think a few words as to whether the results are consistent, and a signpost to the results, would be worth adding.

2. The methods section of the paper describes SAEs by the usual definition, including congenital anomalies and birth defects. Whilst technically not wrong, does this part of the definition really apply in this study?

3. There is nothing I could see about the choice of sample size in the paper, and looking at the protocol in the appendix, there is also nothing to justify the choice of 150 for this first phase of the study. There is a link and reference to the SAP at the start of the methods section, but again, nothing immediately obvious about this. How was this decided upon?

4. The statistical methods used seem good. As a non-Bayesian, there was quite enough in the statistical methods for my taste, but other readers may wish to see more detail. The protocol seems to focus more on the second phase of the study as a whole, and it is only in the SAP that I could find more details. Should this be included in the supplements, and referenced more explicitly in the text of this paper? One thing that I thought was missing from the statistical methods was some information about the choice of prior distributions. I did find this on digging into the SAP, but a quick mention, with reference to the SAP, could be made in the main text.

5. In non-inferiority trials, a justification for the non-inferiority margin is required, and this is provided by reference 18 in the paper. However, when trying to check this, the link did not work for me. (I did get to the document by a different route, and can confirm that the margin chosen is appropriate).

6. Reading the paper, it is not until you reach the end of the discussion that plans for the second phase become clear. It might help, for the sake of context, to describe this earlier on. Note, the paper states a sample size of up to 2000 for the second stage, but the protocol and SAP talk about a maximum of 3000. Is the paper in error?

[LINK] 

General Editorial requests:

1. Please upload figures as individual TIF or EPS files with 300dpi resolution at resubmission; please read our figure guidelines for more information on our requirements: http://journals.plos.org/plosmedicine/s/figures. While revising your submission, please upload your figure files to the PACE digital diagnostic tool, https://pacev2.apexcovantage.com/. PACE helps ensure that figures meet PLOS requirements. To use PACE, you must first registe

---

## [Decision Letter · Decision Letter 2]

22 Apr 2024

Dear Dr. Pérez Chacón,

Thank you very much for re-submitting your revised manuscript (24-00192R2) to PLOS Medicine. The revised manuscript was re-reviewed by a subset of the original reviewers, and as you will see, the reviewers were happy with your responses and all recommend publication. After discussing the paper with the editorial team and the academic editor, I’m pleased to say that we intend to accept the paper for publication, pending some minor editorial issues and any production questions that arise.

The remaining issues that need to be addressed are listed at the end of this email. Any accompanying reviewer attachments can be seen via the link below. Please take these into account before resubmitting your manuscript: [LINK]

When you submit your final revision, please include a point-by-point response to the editorial requests, once again indicating the changes you have made in the manuscript and the corresponding line numbers. Please submit a clean version of the paper as the main article file. A version with changes marked must also be uploaded as a marked up manuscript file.

We ask that you submit your final revision within 1 week if possible. However, if this is not feasible, please contact me directly and we can discuss a suitable alternative (hvanepps@plos.org).

If you have any questions in the meantime, please don’t hesitate to contact me (hvanepps@plos.org). Otherwise, I look forward to seeing your final revision. 

Kind regards,

Heather

Heather Van Epps, PhD

Executive Editor

PLOS Medicine

plosmedicine.org

Comments from Reviewers:

Reviewer #2: 

I want to express my gratitude to the authors for their efforts in incorporating my suggestions, as well as those of the other reviewers and editor

I am pleased to see that the manuscript has undergone significant improvement, and I no longer have any additional concerns.

Reviewer #3:

The authors and satisfactorily addressed my comment, and improved the manuscript. I am happy to recommend acceptance. 

Reviewer #4: 

Alex McConnachie, Statistical Review

I thank the authors for their responses to my original comments, and these are satisfactory. I have no further comments.

Editorial requests:

1. Abstract: please state the primary outcome(s) clearly and specify the analysis population (eg, ITT).

2. Abstract: please include data on the most common AEs and report the 7 SEAs within the findings section of the abstract, per CONSORT for Harms (https://www.equator-network.org/reporting-guidelines/consort-harms/).

3. Abstract: please be sure that the past tense is consistently used, to reflect the fact that phase 1 of the trial is complete (this pertains to the Author summary [‘What did the researchers do and find’ section] as well). 

4. Abstract (minor): please remove the comma in line 63. 

5. Data availability statement: Please remove this paragraph from the final text file (it will be included in the metadata that precedes the published paper). 

6. Methods: lines 170-171: Does this sentence refer to paracetamol (“Two additional doses 6-hour apart were recommended, but not observed by the researchers.”)? Please clarify.

7. Results: please present the data as n (%) of N throughout. For example, at line 273, “72 (96%) of 75 infants in the aP-only schedule group…”

8. Discussion: line 464, please add ‘To our knowledge’ (this should be done wherever there is a claim about primacy). 

9. Figures 2 and 3: please remove ‘boxplots’ from the figure title. Please also add the definitions of ‘aP’ and ‘wP’ to the figure legend.

10. Figures 5-7: please indicate in the figure legends what the number above each bar indicate (n/N).

11. Figures 5 and 6: Please remove the “6 month” heading from the top of the figures, as this is indicated in the figure titles (and is not included in Fig 7).

12. Table 1: please define TAFE in the footnote.

13. References: for relevant refs (URLs) change ‘cited’ to ‘accessed’ and move the accessed date to after the URL; add date accessed for refs 24, 25, 27, 28

14. As a reminder, we ask every co-author listed on the manuscript to fill in a contributing author statement. If any of the co-authors have not filled in the statement, we will remind them to do so when the paper is revised. If all statements are not completed in a timely fashion this could hold up the re-review process. Should there be a problem getting one of your co-authors to fill in a statement we will be in contact. Please do not add or remove authors without first agreeing this with the handling editor. 

[LINK]

---

## [Editor Report · Decision Letter 3]

8 May 2024

Dear Dr Pérez Chacón, 

On behalf of my colleagues and the Academic Editor, James Beeson, I am pleased to inform you that we have agreed to publish your manuscript "Immunogenicity, reactogenicity, and IgE-mediated immune responses of a mixed whole-cell and acellular pertussis vaccine schedule in Australian infants: a randomised, double-blind, non-inferiority trial" (PMEDICINE-D-24-00192R3) in PLOS Medicine.

Many thanks for choosing PLOS Medicine for this work; I look forward to seeing it published. As mentioned in my earlier email, don't hesitate to reach out to me directly if any questions arise during the post-accept process.

Kind regards,

Heather

Heather Van Epps, PhD 

Executive Editor 

PLOS Medicine

hvanepps@plos.org

PRESS
